# A Fast and Selective Approach for Profiling Vicinal Diols Using Liquid Chromatography-Post Column Derivatization-Double Precursor Ion Scanning Mass Spectrometry

**DOI:** 10.3390/molecules27010283

**Published:** 2022-01-03

**Authors:** Debin Wan, Christophe Morisseau, Bruce D. Hammock, Jun Yang

**Affiliations:** Department of Entomology and Nematology and UCD Comprehensive Cancer Center, University of California, One Shields Avenue, Davis, CA 95616, USA; debin.wan@ucdavis.edu (D.W.); chmorisseau@ucdavis.edu (C.M.); bdhammock@ucdavis.edu (B.D.H.)

**Keywords:** vicinal diols, epoxide hydrolase, double precursor ion scan, online post-column derivatization

## Abstract

Vicinal diols are important signaling metabolites of various inflammatory diseases, and some of them are potential biomarkers for some diseases. Utilizing the rapid reaction between diol and 6-bromo-3-pyridinylboronic acid (BPBA), a selective and sensitive approach was established to profile these vicinal diols using liquid chromatography-post column derivatization coupled with double precursor ion scan-mass spectrometry (LC-PCD-DPIS-MS). After derivatization, all BPBA-vicinal-diol esters gave a pair of characteristic isotope ions resulting from ^79^Br and ^81^Br. The unique isotope pattern generated two characteristic fragment ions of *m/z* 200 and 202. Compared to a traditional offline derivatization technique, the new LC-PCD-DPIS-MS method retained the capacity of LC separation. In addition, it is more sensitive and selective than a full scan MS method. As an application, an in vitro study of the metabolism of epoxy fatty acids by human soluble epoxide hydrolase was tested. These vicinal-diol metabolites of individual regioisomers from different types of polyunsaturated fatty acids were easily identified. The limit of detection (LOD) reached as low as 25 nM. The newly developed LC-PCD-DPIS-MS method shows significant advantages in improving the selectivity and therefore can be employed as a powerful tool for profiling vicinal-diol compounds from biological matrices.

## 1. Introduction

Vicinal diols (or 1,2-diols) are a large family of compounds that widely exist in biological systems, such as carbohydrates, nucleosides, nucleotide, and lipids. Among the latter, there are unique and important vicinal-diol metabolites that are generally produced from epoxy fatty acids (EpFAs) by epoxide hydrolases, which are ubiquitous cellular enzymes [1]. Daily dietary intake of polyunsaturated fatty acids (PUFA) is necessary for complete human health. From different PUFAs (linoleic acid (LA, 18:2n6), arachidonic acid (ARA, 20:4n6), dihomo-γ-linolenic acid (DGLA, 20:3n6), α-linolenic acid (ALA, 18:3n3), eicosapentaenoic acid (EPA, 20:5n3), and docosahexaenoic acid (DHA, 22:6n3)), different series of EpFAs are formed by CYP450s [2]. EpFAs regulatory roles in various diseases are becoming clearer, such as anti-inflammatory, analgesic, and normotensive effects [3,4,5,6,7]. EpFAs are regulated in part by being metabolized to vicinal diols mostly by the soluble epoxide hydrolase (sEH) [1]. The sEH is now a therapeutic target for several diseases [8,9,10,11,12,13,14,15]. Unlike EpFAs, the vicinal diols are mostly reported as largely fewer active metabolites of the beneficial epoxides or for their putative toxicity and pro-inflammation effects [16]. Therefore, fatty acid 1,2-diols are potential biomarkers for inflammatory and other diseases. For example, significantly elevated excretion levels of 11,12-DiHETrE and 14,15-DiHETrE, derived from arachidonic acid, were found in patients with pregnancy-induced hypertension [17]. In the urine of patients with peroxisomal diseases, high levels of DiHOME—derived from linoleic acid—glucuronidated conjugates were observed [18], and in this respect, DiHOME could be an important early clinical diagnosis indicator. Previously, studies from our laboratory have demonstrated that DiHOMEs (a.k.a., leukotoxin-diols) are more toxic than the corresponding EpFA (a.k.a., leukotoxin), and leukotoxin-diols are the toxic mediators involved in acute respiratory distress syndrome [19,20]. Recently, 19,20-DiHDPE—derived from DHA—promotes VEGF-induced angiogenesis in mice and its reduction might have additional anti-inflammatory and anti-tumor effects [3]. In addition, abnormally high generation of 19,20-DiHDPE was found in the retinas and vitreous humor of patients with diabetes [21]. Therefore, establishing a rapid and selective method to profile these vicinal diols—present in trace amounts—is essential to discover their biological roles, as well as for establishing new markers of diseases.

The main difficulty in analyzing fatty acid vicinal diols is the presence of numerous regioisomers. For example, four possible regioisomeric diols are formed from ARA—i.e., 5,6-DiHETrE, 8,9-DiHETrE, 11,12-DiHETrE, and 14,15-DiHETrE [22]—which have different biological activities. These regioisomers make a difficult analytical problem even more complicated. So far, liquid chromatography mass spectrometry has proved to be the mostly prominent technique for detection of vicinal-diol compounds [23]. However, it is difficult to separate and quantify by LC-MS all the vicinal diols regioisomeric structures, due to low concentration, poor ionization, and serious matrix interference [22,23]. Pre-column derivatization is an alternative technique to solve these problems. Vicinal-diol compounds react rapidly with boronic acids to generate stable cyclic esters in a neutral or basic media, while the reaction is reversible under acidic conditions [24]. Several boronate affinity techniques were developed to selectively trap vicinal diols prior to instrumental analysis to remove complex co-existing interferences before instrumental analysis [25]. Furthermore, relying on the boronic acid-diol chemistry, various derivatization reagents containing boronic acid have been used to label vicinal-diols to improve MS sensitivity, such as dansyl-3-aminophenylboronic acid, 2-bromopyridine-5-boronic acid, 3-aminophenylboronic acid, 6-methoxy-3-pyridinylboronic acid, 3(4)-(N,N-dimethylamino)-phenylboronic acid, and 4-phenylaminomethyl-benzeneboric acid [26,27,28,29,30,31,32]. However, using pre-column derivatization, the chromatographic behavior of the resulting labeled vicinal-diols compounds is dramatically affected, resulting in a stronger challenge to separate vicinal-diols, especially regioisomeric diols. To solve this, we investigated herein the possibility of post-column derivatization to improve 1,2-diol sensitivity without compromising regioisomers separation.

6-Bromo-3-pyridinylboronic acid (BPBA) was previously reported to label brassinosteroids before instrumental analysis, resulting in a highly sensitive LC-MS method [27,33]. Based on the fast and easy reaction between BPBA and vicinal diols, we choose BPBA as the labeling reagent to establish a liquid chromatography post column derivatization double precursor ion scan-mass spectrometry (LC-PCD-DPIS-MS) to identify selectively vicinal-diols metabolites of PUFAs. In this approach, the vicinal diol regioisomers are already separated from each other, and, in addition, the derivatization reaction should not be affected by the sample matrix because it should be separated from the targeted diols. Finally, the isotope distribution of bromine can be used as an isotope labeling; the double fragment ions resulting from native isotope distribution of ^79^Br and ^81^Br enabled DPIS-MS, which should facilitate the identification of vicinal-diol compounds. To demonstrate the usefulness of the optimized novel method, it was used to profile the products formed by the human sEH when exposed to a mixture of EpFAs resembling the in vivo situation.

## 2. Results and Discussion

### 2.1. Mass Spectrometry Behavior of BPBA Labeled Compounds

BPBA, a boronic acid containing reagent, should react quickly with vicinal diols to form stable boronic esters [27,33]. Therefore, to test the reaction speed 14,15-DiHETrE was initially chosen as a representative vicinal diol to optimize the derivatization reaction off-line (Appendix A). 100 μL of a 1μM solution of 14,15-DiHETrE in acetonitrile was mixed with 100 μL of 1 mM BPBA solution in acetonitrile. The mixture was then directly injected into a mass spectrometer using a syringe pump for analysis. A Waters LCT was used to initially acquire the full-scan mass spectrometric analysis of BPBA derivatized 14,15-DiHETrE, because it is well-known that a time-of-flight mass analyzer (TOF-MS) is of higher resolution and relatively higher sensitivity than quadrupole mass analyzer under full scan mode. Full-scan mass spectrometric analysis of BPBA derivatized 14,15-DiHETrE is presented in Figure 1.

The BPBA-14,15-DiHETrE ester shows two peaks with *m/z* at 502 and 504 with a 1:1 ratio, forming from the isotope distribution of bromine, ^79^Br and ^81^Br. There is no visible peak corresponding to 14,15-DiHETrE or the mono hydroxyl ester of 14,15-DiHETrE. The two peaks with *m/z* at 200 and 202 are the ions of BPBA. In addition, the peaks with central *m/z* at 384.90 and *m/z* at 567.85 are dimer and trimer of BPBA, respectively, due to the high concentration of BPBA. The product ion scan of BPBA derivatized 14,15-DiHETrE (*m/z* at 502 and 504) was obtained on an AB Sciex 4000 Qtrap. The tandem mass spectra of these twin peaks of BPBA derivatized 14,15-DiHETrE are shown in Figure 2A,B, respectively. The major product ions are peaks with *m/z* at 200 and 202, corresponding to [BPBA (^79^Br)–H]^−^ and [BPBA (^81^Br)–H]^−^, respectively. The minor product ions with *m/z* at 78.8 and 81.0 are corresponding to [^79^Br–H]^−^ and [^81^Br–H]^−^, respectively.

### 2.2. Construction of PCD and Optimization

The vicinal diols reaction with BPBA is very rapid (few seconds), and therefore the post column derivatization (PCD) was carried out with the configuration as described in Appendix A. The BPBA solution was delivered with a syringe pump into the LC eluate; the derivatization reaction took place in a zero dead-volume cross tee, used to avoid peak broadening, which is commonly seen in PCD combined with LC-MS detection. To optimize the reaction, a modifier solution was propelled using another syringe pump into the tee connector. Different organic solvents, concentrations of BPBA, flow rates, and reaction modifiers were optimized to obtain the highest signal of diol-BPBA derivatives. Methanol and acetonitrile are commonly used as LC mobile phase. Because methanol could react with BPBA, acetonitrile was thus employed as the solvent to prepare the BPBA solution. The effects of different concentrations of BPBA solution on the derivatization were also evaluated (Figure 3). The highest signal response was obtained with a solution of 100 μM of BPBA (Figure 3A). Further analysis of the product-to-reactant signal ratios, the highest relative yield was obtained with 200 μM of BPBA (Figure 3B). This suggests that the ionization of the derivatized product might be inhibited by BPBA when its concentration is over 100 μM. This observation was confirmed by testing the signal intensity of CUDA in the presence of different concentrations of BPBA solutions (data not shown). This inhibition also explains the high standard deviations (~20% as shown in Figure 3A,B) when using over 100 µM of BPBA solutions. Therefore, 100 μM of BPBA was chosen as the optimal amount of derivatization reagent. Next, the flow rate (5, 10, and 20 μL/min) of pumping the BPBA solution was also studied. The lowest flow rate tested (5 μL/min) resulted in an unstable signal of the derivatives. Both 10 and 20 μL/min pumping rates gave a stable signal for the products. Thus, to avoid high consumption the BPBA solution, 10 μL/min was chosen as the pumping rate. A basic reaction medium was reported to be helpful for the reaction between boronic acid and diols. However, based on the product signal intensity, we observed the strongest response upon using a neutral rather than modifier solution (Appendix A). This may be due to a decrease in the ionization efficiency of the products when the basic solution was introduced directly before the ion source. Because the LC solvent is already neutral, no modifier solution was used in the final configuration. Put together, the optimized conditions for the PCD are 100 μM of BPBA in acetonitrile provided by a syringe pump at a rate of 10 μL/min. After optimization, the yield of the derivatization reaction is >75% and stable.

### 2.3. Double Precursor Ion Scan

Some published methods [34] report the use of deuterated reagent to permit the identification of unknown compounds following a reaction similar to the one used here. However, compared to other boronic acid reagents available, BPBA has the unusual advantage of containing a bromine atom yielding its characteristic isotopic distribution pattern derived from bromine, which facilitates the identification of diols in complex samples. The BPBA derivatives have a characteristic 1:1 isotope pattern resulting from ^79^Br and ^81^Br, so their paired peaks at the exact same retention time are ideal to selectively detect vicinal-diols compounds in complex samples. As described above, the pair of ions *m/z* 200 and 202 are the dominant fragment ions in MS/MS analysis. Therefore, precursor ion scans (DPIS) of *m/z* at 200 and 202 can be helpful in identifying vicinal diols. Peak-pair data can be extracted from these two ion chromatograms (*m/z* at 200 and 202). Only these peak pairs with the same retention time and intensities are assigned to arise from vicinal diol derivatives. To ensure that the DPIS method is working well for other potential vicinal diols, several vicinal diols from different fatty acids were randomly selected to react with BPBA. Their fragmentation patterns were investigated. As expected, similar to the fragmentation pattern of BPBA 14,15-DiHETrE ester, the product ions with *m/z* at 200 and 202 are the major peaks for all the vicinal diols tested. This underlines the suitability of the DPIS method developed herein to detect all potential vicinal diols.

In precursor ion scan mode, the CID energy is a key factor that affects sensitivity significantly. The optimized CID energy for each compound is shown in Appendix A. Interestingly, the different chain lengths and different saturations of the fatty acids do not affect the optimum collision energy of the BPBA-vicinal-diols esters significantly. Therefore, the CID energy was set at −33 eV.

### 2.4. Optimization of Chromatographic Separation

Because methanol could react with BPBA, acetonitrile was chosen as the organic mobile phase for separation. In addition, an acid solution could decrease the derivatization reaction yield, therefore no acid was added to any of the mobile phases. Thus, mobile phase A and phase B are pure neutral deionized water and acetonitrile, respectively. The LC conditions, including optimized column temperature, flow rate, and gradient were optimized to separate the regioisomeric vicinal diols from several commons unsaturated fatty acids before the BPBA labeling. The final chromatograms are shown in Figure 4. All these regioisomeric vicinal diols derived from each EpFA can reach baseline separation before PCD.

### 2.5. Comparison of Pre-Column Derivatization LC-FMS and LC-PCD-DPIS-MS

Liquid chromatography (LC) combined with full-scan mass spectrometry (FMS) and tandem mass spectrometry (MS/MS) is recognized as a classical method to identify chemicals. To increase MS sensitivity, offline derivatization is often used. Here, the developed LC-PCD-DPIS-MS was compared with traditional offline pre-instrument derivatization combined with LC-FMS. 500 nM of 14,15-DiHETrE was spiked in a 10-fold diluted pooled human serum. After extraction, the residue was divided in two parts. The first half was derivatized offline with BPBA then injected into the LC for the pre-column derivatization LC-FMS approach. The other half sample was directly analyzed by LC-PCD-DPIS-MS method described above on the same AB Sciex 4000 Qtrap. After derivatization, properties of analytes were changed significantly. The clogP value of BPBA-14,15-DiHETrE ester is 7.15 (ChemDraw, Perkin Elmer, Waltham, MA, USA), which is much higher than that of 14,15-DiHETrE (4.31). Consequently, the BPBA-14,15-DiHETrE ester adheres strongly to the C18 column. It was not possible to elute the esters out on the same Eclipse Plus C18 column with the solvent system used. Therefore, we used a short column with different solvent system for analyzing the offline derivatives (elution condition stated in Appendix A). Figure 5A shows the extracted ion chromatograms of BPBA (^79^Br) 14,15-DiHETrE ester (*m/z* = 502) of LC-FMS. The signal-to-noise results of BPBA (^79^Br) 14,15-DiHETrE ester is 81.6. As shown above, it is possible to separate the vicinal diols on the C18 column. Figure 5B presents the extracted ion chromatograms of *m/z* 200 that were acquired by LC-PCD-DPIS-MS. Compared to the result obtained by PID-LC-FMS, the signal of 14,15-DiHETrE is quite clear, and S/N is 499.9. The LOD of BPBA-14,15-DiHETrE ester using LC-PDC-DPIS-MS was determined to be 25 nM. These results underline the advantage of post-column derivatization because it does not affect previously optimized separation of the vicinal-diol isomers or other analytes of interest. In addition, the derivatization reaction is unaffected by the sample matrix since the targeted analytes have been already separated from the possible existing interference before derivatization.

### 2.6. Reproducibility of the Developed LC-PCD-DPIS-MS Method

The robustness of the method is critical although the described method was developed for discovering new vicinal diols in the biological matrices. We designed a reproducibility experiment to assess the robustness of the developed LC-PCD-DPIS-MS method with the three different concentrations of 14,15 DiHETrE (50, 500, and 5000 nM) with 500 nM of d11 11,12 DiHETrE as the internal standard. The relative standard deviations of intraday and inter-day were given in Appendix A. They have all shown very promising reproducibility (RSD < 15%).

### 2.7. In Vitro Study of HsEH Metabolites of EpFA

EpFAs are very important signaling mediators in a variety of biological processes The sEH, expressed in almost every organ, is the primary enzyme that degrades these chemicals in vivo, by degrading the EpFAs to their corresponding and usually less-active vicinal diols [1]. The pharmacological inhibition of sEH for stabilizing EpFAs is considered for the treatment of numerous diseases [8,9,10,11,12,13,14,15]. Using purified EpFAs regioisomers, it was shown that sEH has selectivity among the different regioisomers of these natural epoxides [35,36]. However, there is a mixture of EpFAs, in terms of fatty acid backbone, but also in terms of regioisomers, present in the cells. Given the different biology associated with these different EpFAs, it is important to understand which ones are metabolized faster by sEH within a mixture of these compounds.

To test this, and to illustrate the value of the method described to detect all possible vicinal diols derived from these different series of EpFAs, a mixture of EpFAs was partially hydrolyzed by the human sEH. The resulting diols were extracted and analyzed by LC-PCD-DPIS-MS. As shown in Figure 6, numerous vicinal diols were clearly identified—i.e., 17,18-DiHETE, 14,15-DiHETE, 11,12-DiHETE and 8,9-DiHETE derived from EPA; 12,13-DiHOME and 9,10-DiHOME derived from linoleic acid; 19,20-DiHDPE, 16,17-DiHDPE, 13,14-DiHDPE, 10,11-DiHDPE, and 7,8-DiHDPE from DHA; 14,15-DiHETrE, 11,12-DiHETrE, and 8,9-DiHETrE from ARA. Interestingly, 5,6-DiHETE, 4,5-DiHDPE, and 5,6-DiHETrE were not detectable in these samples, indicating their corresponding EpFAs are probably hydrolyzed much slower than other EpFAs by the human sEH. This result is consistent with previously reported in vitro rates of hydrolysis for the human sEH, for which the epoxides closest to the carboxylic acid are hydrolyzed at the lowest velocity [35,36].

## 3. Materials and Methods

### 3.1. Chemicals and Reagents

All tested *cis*-diol standards, dihydroxyoctamonoenoic acids (DiHOMEs), including 9,10-DiHOME and 12,13-DiHOME); dihydroxyeicosatrienoic acids (DiHETrEs), including 14,15-DiHETrE, 11,12-DiHETrE, 8,9-DiHETrE and 5,6-DiHETrE; dihydroxyoctadecadienoic acids (DiHODEs), including 15,16-DiHODE, 12,13-DiHODE, 9,10-DiHODE and 7,8-DiHODE; dihydroxyeicosatetraenoic acids (DiHETEs), including 17,18-DiHETE, 14,15-DiHETE, 11,12-DiHETE, 8,9-DiHETE, and 5,6-DiHETE; and dihydroxydocosapentaenoic acids (DiHDPEs), including 19,20-DiHDPE, 16,17-DiHDPE, 13,14-DiHDPE, 10,11-DiHDPE, 7,8-DiHDPE, and 4,5-DiHDPE and 14,15-EpETrE, 11,12-EpETrE, 8,9-EpETrE and 5,6-EpETrE from ARA; 17,18-EpETE, 14,15-EpETE, 11,12-EpETE, 8,9-EpETE, and 5,6-EpETE from EPA were purchased from Cayman Chemical (Ann Arbor, MI, USA) and Biomol Research laboratories, Inc. (Plymouth Meeting, PA, USA). 1-cyclohexyl-dodecanoic acid urea (CUDA) was synthesized previously in our laboratory [37]. Human soluble epoxide hydrolase (human sEH) was expressed and purified as described [38]. All standards were stored under −80 °C until actively utilized for analysis. Oasis HLB 60 mg SPE cartridges were purchased from Waters Co. (Milford, MA, USA). Acetonitrile and methanol of LC-MS grade were purchased from Thermo Fisher (Pittsburgh, PA, USA). 6-Bromo-3-pyridinylboronic acid (BPBA) and all other chemical reagents were purchased from Sigma (St. Louis, MO, USA).

### 3.2. Sampling

The bioactive lipid vicinal diols in pooled human plasma (Sigma Aldrich, St Louis, MO, USA) and cell culture fluid were purified by following our previously described protocol [39]. Briefly, deuterated *cis*-diols, d11-14,15-DiHETrE were spiked into these samples to mimic the *cis*-diols, and then these samples were enriched by SPE extraction. The collected samples were dried in a vacuum centrifuge, and 50 μL of 200 nM CUDA in methanol was added to reconstruct the analyte solution. CUDA here was used to verify the potential ionization variability which could occur in the ion source.

### 3.3. Chromatography and Post-Column Derivatization

The on-line post-column derivatization (PCD) system was constructed in house (shown in Appendix A). The separation of analytes was performed on an Agilent 1200 SL liquid chromatography system (Agilent Corporation, Palo Alto, CA, USA). The auto-sampler was kept at 10 °C. The column used for separation was an Agilent ZORBAX Eclipse Plus C18, 1.8 µm, 2.1 × 150 mm and the temperature was kept at 50 °C. Mobile phase A and B were pure deionized water and acetonitrile, respectively. The flow rate was set as 250 μL/min. The gradient was optimized and is given in Appendix A. In short, the gradient starts from 35% of acetonitrile for 0.25 min, then increases to 45% at 1 min, 55% at 3 min, 66% at 8.5 min, 72% at 12.5 min, 82% at 15 min, holds at 95% from 16.5 to 18 min, then is retained 35% in the end from 18.1 to 21.5 min. The injection volume is 10 µL. The syringe pump (KD Scientific KDS100, Holliston, MA, USA) supplied 100 μM of BPBA solution in acetonitrile with a flow rate at 10 μL/min.

### 3.4. Mass Spectrometry

High-resolution mass spectra were acquired on Micromass LCT (Micromass, Manchester, UK), an orthogonal acceleration-time-of-flight (oa-TOF) mass spectrometer with electrospray ionization (ESI) source. The instrument was operated in negative ion mode. The lock-spray source used leucine-enkephalin as reference ions independently.

The tandem mass spectrometric analysis was performed on a 4000 Qtrap tandem mass spectrometer equipped with a Turbo V^®^ electrospray source. The instrument was operated in two different negative modes, full scan and double precursor ion scan (DPIS). DPIS was carried out in negative ion mode. The DPIS method consisted of two precursor ion scans—i.e., ions at *m/z* 200 and 202. The ESI source parameters were set up as shown in Appendix A.

### 3.5. Determination of Metabolites of HsEH

0.1 mM of each of epoxy fatty acids (EpFA) was prepared in DMSO including EpOMEs (12,13-EpOME and 9,10-EpOME) derived from LA; EpETrEs (14,15-EpETrE, 11,12-EpETrE, 8,9-EpETrE, and 5,6-EpETrE) derived from ARA; EpETEs (17,18-EpETE, 14,15-EpETE, 11,12-EpETE, 8,9-EpETE, and 5,6-EpETE) derived from EPA; and EpDPEs (19,20-EpDPE, 16,17-EpDPE, 13,14-EpDPE, 10,11-EpDPE, and 7,8-EpDPE) from DHA. 0.4 μg/mL of purified HsEH was prepared in 100 uL of sodium phosphate buffer (0.1 M, pH 7.4) containing 0.1 mg/mL of BSA. The EpFA mixture with each compound at a final concentration of 1 μm was incubated with HsEH, and incubated at 30 °C for 20 min. The enzyme action was quenched by addition of 100 μL methanol. After centrifugation, the potential vicinal-diols metabolites in supernatant were identified by LC-PCD-DPIS-MS.

## 4. Conclusions

This paper demonstrates the efficacy and usefulness of a new LC-PCD-DPIS-MS method for identifyingvicinal diols. The method is more selective and sensitive than the traditional precolumn derivatization LC-FMS method. The BPBA-PCD combined with LC-MS/MS method shows several advantages as follows: (1) Due to the characteristic 1:1 isotope distribution of bromine (^79^Br and ^81^Br) of BPBA-vicinal-diols ester, the extracted ion chromatograms of *m/z* 200 and *m/z* 202 under DPIS shows these peaks at the same retention time, which is easier and more confident than DPIS using both a native and deuterated labeling reagent. (2) The physical properties of the analytes do not change during chromatography; thus, separation of potential targeted vicinal diols can be easily achieved. (3) Since the existing interference that may react with BPBA have already been separated from the targeted analytes, the derivatization reaction of vicinal diols will not be affected by matrix. Using this developed strategy, the metabolic selectivity of the human sEH enzyme was detected. According to the achieved results, it is suggested that the developed LC-PCD-DPIS-MS method could be effectively applied for the routine profiling analysis of vicinal-diol containing compounds. The developed LC-PCD-DPIS-MS assay displays far-superior sensitivity and specificity than either conventional LC-Full scan MS (LC-FMS) and pre-instrument derivatization (PID)-LC-FMS, and it is therefore very powerful for discovery of new vicinal diol contained compounds. 

It is worth noting that the current developed method is for discovering and profiling the new vicinal diol contained compounds in the biological matrices instead of the quantification of them. For the quantification of the diols, the selected reaction monitor (SRM) mode of the triple quadrupole mass spectrometer will be more sensitive and a better choice. The SRM methods for these diols should be straightforward to develop after their identification.

## Figures and Tables

**Figure 1 molecules-27-00283-f001:**
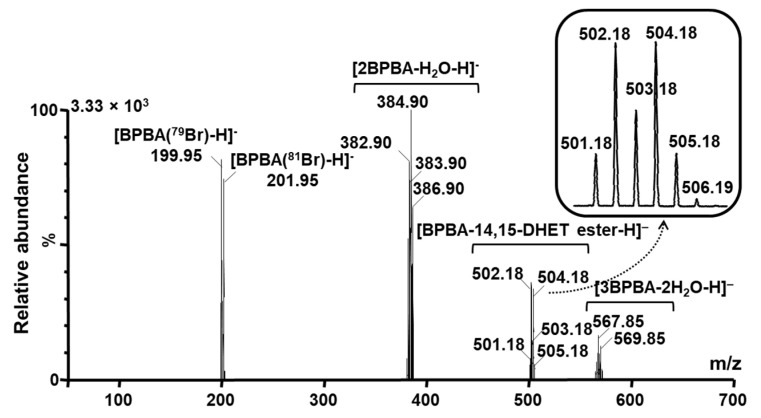
Full scan mass spectrum of the BPBA labeled 14,15-DiHETrE using Micromass LCT in negative ion mode.

**Figure 2 molecules-27-00283-f002:**
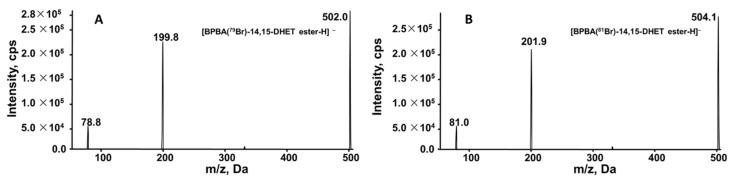
Tandem mass spectra of BPBA labeled 14,15-DiHETrE. (**A**) BPBA (^79^Br) 14,15-DiHETrE ester; (**B**) BPBA (^81^Br) 14,15-DiHETrE ester.

**Figure 3 molecules-27-00283-f003:**
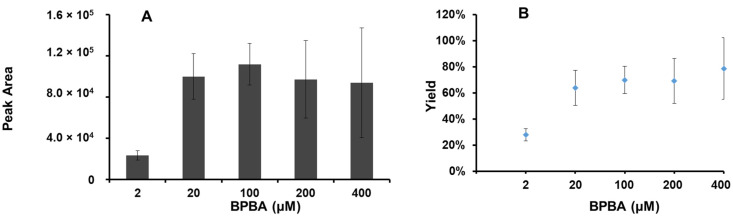
Optimization of post column derivatization reaction using different concentration of BPBA. (**A**) Signal of BPBA labeled 14,15-DiHETrE; (**B**) Yield of BPBA labeled 14,15-DiHETrE.

**Figure 4 molecules-27-00283-f004:**
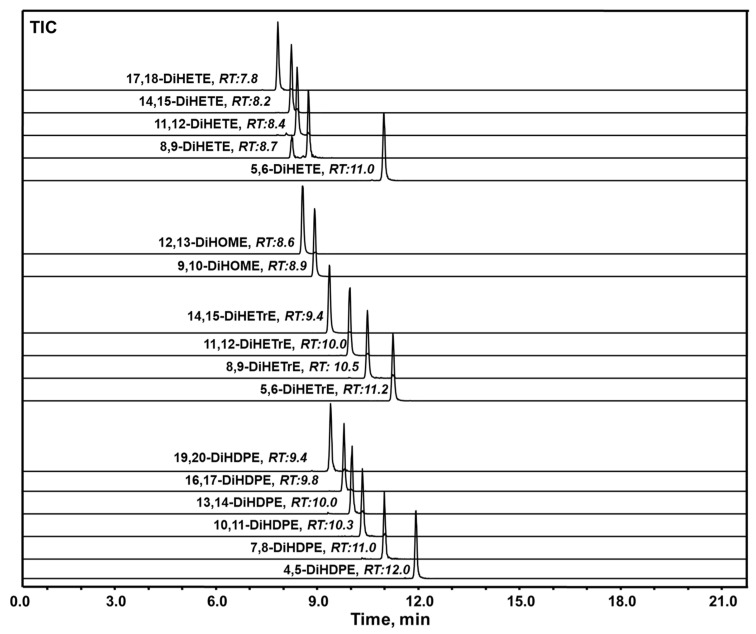
Chromatography of vicinal cis-diol metabolites generated from EpFAs. The separation condition is described in the experimental section. The HPLC gradient for separation of vicinal diols is provided in Appendix A.

**Figure 5 molecules-27-00283-f005:**
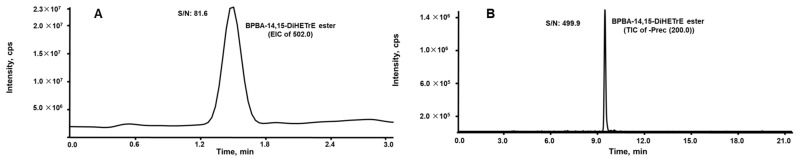
Comparison of offline derivatization LC-FMS and LC-PCD-DPIS-MS methods for detection of 14,15-DiHETrE. (**A**) Extracted ion chromatogram of BPBA (^81^Br) 14,15-DiHETrE ester. (**B**) Chromatogram of PIS of *m/z* 200.

**Figure 6 molecules-27-00283-f006:**
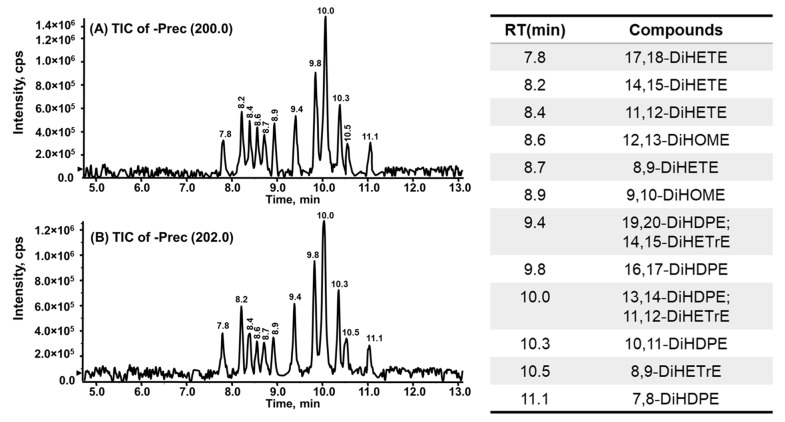
Identification of vicinal diols formed from human sEH hydration of a mixture of EpFAs, which presented in both precursor scan of *m/z* 200 (**A**) and 202 (**B**).

## Data Availability

All the data supporting this study are provided here or in the Appendix A document.

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
