# Peer review of "A Fast and Selective Approach for Profiling Vicinal Diols Using Liquid Chromatography-Post Column Derivatization-Double Precursor Ion Scanning Mass Spectrometry"

_molecules, 2022, doi:10.3390/molecules27010283_

Round 1
Reviewer 1 Report
Although the topic is of interest, the manuscript suffers from some rather important deficiencies. Below please find some specific comments:
Line 124: correct to "reaction"
Line 127: replace "happened" with "took place"
Line 129: replace "given" with "propelled"
Line 144: replace "wasting" with "high consumption"
Line 145: rephrase to "basic reaction medium"
Figure 3A&B: please comment of the very high standard deviations at 200 and 400 μM of the derivatizing reagent.
Lines 157 and 158: please provide references
Section 2.5: "pre column derivatization" is a rather more suitable term compared to "pre-instrument"
Section 2.5: the authors compare pre and post column derivatization using the same LC conditions; the fact that pre-column derivatives could not be eluted should have been expected. Different LC conditions should have been used for a fully justified comparison.
Section 3.3: the manufacturer of the syringe pump is missing.
Validation: validation experiments/data are almost totally missing; accuracy, precision, matrix effect etc.
Author Response
Reviewer #1:
Although the topic is of interest, the manuscript suffers from some rather important deficiencies. Below please find some specific comments:
- Line 124: correct to "reaction"
Changed as suggested.
- Line 127: replace "happened" with "took place"
Changed as suggested.
- Line 129: replace "given" with "propelled"
Changed as suggested.
- Line 144: replace "wasting" with "high consumption"
Changed as suggested.
- Line 145: rephrase to "basic reaction medium"
Changed as suggested.
- Figure 3A&B: please comment of the very high standard deviations at 200 and 400 μM of the derivatizing reagent.
This is a great point. The competition of ionization is the main reason for the high standard deviations at 200 and 400 µM of BPBA. We added the comments line 137-142 as copied below:
“This suggests that the ionization of the derivatized product might be inhibited by BPBA when its concentration is over 100 μM. This observation was confirmed by testing the signal intensity of CUDA in the presence of different concentrations of BPBA solutions (data not shown). This inhibition also explains the high standard deviations (~20% as shown in Figure 3A and 3B) when using over 100 µM BPBA solutions.”
- Lines 157 and 158: please provide references
Changed as suggested. The reference was added as [34] listed in the end.
- Section 2.5: "pre column derivatization" is a rather more suitable term compared to "pre-instrument"
Changed as suggested.
- Section 2.5: the authors compare pre and post column derivatization using the same LC conditions; the fact that pre-column derivatives could not be eluted should have been expected. Different LC conditions should have been used for a fully justified comparison.
This is a great point which we fully agree with. In fact, in order to compare with it, we were using a short column to elute the pre-column derivatives as shown in Figure 5A. We realized that we did not make it clear in the text to reflect that. We added several sentences now from line 210-213 to clarify that:
“It was not possible to elute the esters out on the same Eclipse Plus C18 column with the solvent system used. Therefore, we used a short column with different solvent system for analyzing the offline derivatives (elution condition stated in supplementary information).”
- Section 3.3: the manufacturer of the syringe pump is missing.
Changed as suggested. We added the details on line 297-298.
“The syringe pump (KD Scientific KDS100, Holliston, MA USA) supplied 100μM BPBA solution in acetonitrile with a flow rate at 10μL/min.”
- Validation: validation experiments/data are almost totally missing; accuracy, precision, matrix effect etc.
This is a great point, and we totally agree that full validation experiments/data are needed if the reported method was developed for quantifying vicinal diols in the biological samples. However, the current method we developed is not for quantification of these vicinal diols. Instead, this method is designed for discovering new vicinal diol contained compounds as we stated in both introduction (line 54-56):
“Therefore, establishing a rapid and selective method to detect these vicinal diols, present in trace amounts, is essential to discover their biological roles as well as for establishing new markers of diseases.”
And conclusion (line 337-340):
“The developed LC-PCD-DPIS-MS assay displays far-superior sensitivity and specificity than either conventional LC-Full scan MS (LC-FMS) and pre-instrument derivatization (PID)-LC-FMS, and it is therefore very powerful for discovery of new vicinal diol contained compounds.”
For the quantification of the diols, the selected reaction monitor (SRM) mode of the triple quadrupole mass spectrometer will be much sensitive and better choice. And the SRM methods for these diols should be straightforward to develop after identification of them. In order to convey this information and clarify this point, we added the sentences at the end of the conclusion part as below:
“It’s worth noting that the current developed method is for discovering the new vicinal diol contained compounds in the biological matrices instead of the quantification of them. For the quantification of the diols, the selected reaction monitor (SRM) mode of the triple quadrupole mass spectrometer will be much sensitive and better choice. And the SRM methods for these diols should be straightforward to develop after identification of them.”
We agree that it is still very critical to have the reproducibility data for the method to ensure the robustness of the developed method. We added a whole section as the new section 2.6. The old section 2.6 was changed to section 2.7 now.
Reviewer 2 Report
This manuscript reports the application of a combined post-column derivatization and LC-MS method using 6-bromo-3-pyridinylboronic acid (BPBA) as a derivatizing reagent to vicinal diols. However, it is difficult to separate them due to the presence of many stereoisomers. The post-column derivatization method solves the problems of separation and sensitivity, and the experiments conducted in this paper are very interesting. However, there are some points that need to be resolved before this manuscript can be accepted. These are as follows.
1. The post-column derivatization method has been reported for a long time. The reason for this is that the non-volatile derivatization reagents added in the post-column cannot be removed, and the accumulated derivatization reagents are considered to be a burden on LC-MS. There is only one report of a post-column derivatization method using boronic acid-diol chemistry as far as I could find. (https://doi.org/10.1021/acs.jafc.9b07623) Therefore, the authors need to investigate the removal of non-volatile derivatization reagents and their effect on LC-MS.
2. The reproducibility of the developed method has not been tested much, but the error bars in figure 3 seem to indicate a measurement variability of more than 20%. In order to perform quantitative analysis, please use an internal standard and perform correction.
3. The validity and robustness of the developed method have not been investigated much. Please refer to the following FDA guideline and conduct additional experiments for method validation.
https://www.fda.gov/regulatory-information/search-fda-guidance-documents/bioanalytical-method-validation-guidance-industry
Comparison of pre-instrument derivatization (PID)-LC-FMS and LC-PCD-DPIS-MS in Session 2.5 is of little value. If the model compound does not elute from the column, then you need to select a model compound that does elute from the column. In addition, the measurement mode of MS must be unified to be able to compare them.
Author Response
Reviewer #2:
This manuscript reports the application of a combined post-column derivatization and LC-MS method using 6-bromo-3-pyridinylboronic acid (BPBA) as a derivatizing reagent to vicinal diols. However, it is difficult to separate them due to the presence of many stereoisomers. The post-column derivatization method solves the problems of separation and sensitivity, and the experiments conducted in this paper are very interesting. However, there are some points that need to be resolved before this manuscript can be accepted. These are as follows.
- The post-column derivatization method has been reported for a long time. The reason for this is that the non-volatile derivatization reagents added in the post-column cannot be removed, and the accumulated derivatization reagents are considered to be a burden on LC-MS. There is only one report of a post-column derivatization method using boronic acid-diol chemistry as far as I could find. (https://doi.org/10.1021/acs.jafc.9b07623) Therefore, the authors need to investigate the removal of non-volatile derivatization reagents and their effect on LC-MS.
This is a great point and very thoughtful. We agree that the poor volatility is a big limitation for the usage of BPBA. However, this is not a huge issue for the current project since the current method is for discovery study of vicinal diols in biological matrices. In our application, the method will not serve as a routine quantitation assay for a biochemical reaction. Therefore, we will not have huge amounts of derivatization reagents injected into the mass spectrometer. In addition, the final amount of the derivatization reagents injected into the mass spectrometer was diluted by 26 times (10 uL/min flow combine with 250 uL/min flow from pump). Because of these two reasons, in our application, we have not observed severe adverse effects on the mass spectrometer yet.
- The reproducibility of the developed method has not been tested much, but the error bars in figure 3 seem to indicate a measurement variability of more than 20%. In order to perform quantitative analysis, please use an internal standard and perform correction.
This is a great point. As we replied in Reviewer #1’s comment (#11), the current method was not developed for quantitative analysis. We revised our manuscript to emphasize and clarify this as shown in revision and the reply to Reviewer #1’s comment.
Although our method was not developed for quantitative analyses, we do valuate the reproducibility of the developed method. Therefore, we added experiments of the reproducibility and followed the reviewer’s suggestion by adding the internal standard (d11 11,12 DiHETrE) to perform the correction. The result is given in section 2.6 and Table S4 now.
- The validity and robustness of the developed method have not been investigated much. Please refer to the following FDA guideline and conduct additional experiments for method validation.
https://www.fda.gov/regulatory-information/search-fda-guidance-documents/bioanalytical-method-validation-guidance-industry
As we replied to Reviewer 1 comment 11 and Reviewer 2 comment 13, the current method was not for quantitative analyses. We feel the full validation of the method is not necessary. But we do agree that the robustness of the method is essential even for the discovery work we are targeting on. Therefore, we added a whole section 2.6 to assess the reproducibility of the developed method. The result is given in section 2.6 and Table s2 now.
- Comparison of pre-instrument derivatization (PID)-LC-FMS and LC-PCD-DPIS-MS in Session 2.5 is of little value. If the model compound does not elute from the column, then you need to select a model compound that does elute from the column. In addition, the measurement mode of MS must be unified to be able to compare them.
This is a great point. We realize that we did not make it clear in the original manuscript. In fact, to compare PID-LC-MS method with LC-PCD-DPIS-MS, we were using a short column to elute the pre-column derivatives as shown in Figure 5A. We added several sentences in the current revision from line 210-213 to clarify that:
“It was not possible to elute the esters out on the same Eclipse Plus C18 column with the solvent system used. Therefore, we used a short column with different solvent system for analyzing the offline derivatives (elution condition stated in supplementary information).”
The niche of the current developed method was to utilize the characteristic isotope of Br to perform precursor scans of m/z 200 and m/z 202, which improves the selectivity of the current method to discover the unknown vicinal diols in the samples. The authors feel that the comparison with full scan method is proper here.
Reviewer 3 Report
The paper describes an original HPLC-MS method for determining several fatty acid 1,2-diols based on post-column derivation of vicinal diols with 6-bromo-3-pyridinylboronic acid (BPBA). The importance of this study, the choice of reagent and derivatization approach (post-column) are clearly explained by authors, and its content and quality are very favorable for its publication in Molecules. There are a few observations related to the content of this manuscript, which require some clarifications, such as:
In sections 2.2 and 2.4 it was emphasized that the choice of acetonitrile was based on the fact that methanol could react with BPBA, but the mobile phase is based on acetonitrile/methanol (84:16), section 3.3.
The significant difference between auto-sampler set-up temperature (10 C) and column temperature (50 C) is surprising.
The gradient elution program mentioned in Table S1 should be included in section 3.3.
Injection volume should be indicated in section 3.3.
Methodology of calculating log P for some ester species should be indicated (section 2.5).
A schema for the derivatization reaction does not dilute the paper content and could give a better image of the derivatization procedure.
It is unclear why (Fig. 2), for example, the ratio between line intensity of m/z = 199.8, derived from the high excess of BPBA (1000 times higher than 14,15-DiHETrE in acetonitrile; see lines 98, pag. 3) is much lower than m/z = 502 (ester); it is supposed that both the reagent excess and derivative product (ester) are simultaneously participating to the MS process. This seems in contradiction with explication given in section 2.2, when <the ionization of the derivatized product might be inhibited by BPBA when its concentration is over 100 uM>.
Author Response
Reviewer #3:
The paper describes an original HPLC-MS method for determining several fatty acid 1,2-diols based on post-column derivation of vicinal diols with 6-bromo-3-pyridinylboronic acid (BPBA). The importance of this study, the choice of reagent and derivatization approach (post-column) are clearly explained by authors, and its content and quality are very favorable for its publication in Molecules. There are a few observations related to the content of this manuscript, which require some clarifications, such as:
- In sections 2.2 and 2.4 it was emphasized that the choice of acetonitrile was based on the fact that methanol could react with BPBA, but the mobile phase is based on acetonitrile/methanol (84:16), section 3.3.
A great catch. It’s a mistake in the method section 3.3. The phase B is acetonitrile without methanol. We have fixed it in the current revision. (line 296).
- The significant difference between auto-sampler set-up temperature (10 C) and column temperature (50 C) is surprising.
The low temperature setting (10 ℃) in autosampler was due to the potential concern of the stability of the target diols. The relative high temperature (50 ℃) setting for column oven is to reduce the back pressure on the UPLC column.
- The gradient elution program mentioned in Table S1 should be included in section 3.3. Injection volume should be indicated in section 3.3.
Changed as suggested.
- Methodology of calculating log P for some ester species should be indicated (section 2.5).
The ClogP stated in the manuscript is according to the calculation in ChemDraw software. We added the details in the revision (line 208).
- A schema for the derivatization reaction does not dilute the paper content and could give a better image of the derivatization procedure.
Changed as suggested.
- It is unclear why (Fig. 2), for example, the ratio between line intensity of m/z = 199.8, derived from the high excess of BPBA (1000 times higher than 14,15-DiHETrE in acetonitrile; see lines 98, page. 3) is much lower than m/z = 502 (ester); it is supposed that both the reagent excess and derivative product (ester) are simultaneously participating to the MS process. This seems in contradiction with explication given in section 2.2, when <the ionization of the derivatized product might be inhibited by BPBA when its concentration is over 100 uM>.
We are sorry for the confusion. Figure 2 shows the tandem mass spectra of the derivative product (ester). So, in the spectra, m/z of 199.8 is only from the fragmentation of the ester instead of from the derivatization reagent.
On the contrary, Figure 1 shows the full scan mass spectrum, which is consistent of the reviewer’s expectation that the line intensity of BPBA (m/z 199.95, excess amount) is higher than its product (m/z 502.18).
Round 2
Reviewer 1 Report
The revised version can be accepted for publication
Author Response
Thanks a lot! Reviewer 1's comments really helped us to improve the manuscript a lot! We really appreciate them.
Reviewer 2 Report
As for the resubmitted manuscript, it has been revised in some parts but not completely.
The authors complain that the purpose of this manuscript is not quantitative analysis, otherwise the purpose of this manuscript is unclear.
I think the purpose of this manuscript needs to be clarified and the structure of the manuscript needs to be reconsidered.
Therefore, I have judged that this manuscript has not been sufficiently improved to warrant publication in Molecules.
Author Response
Changed as requested.
We understand that some future readers might share the same confusion as Reviewer 2 about the purpose of the developed method. In the current revision, we tried our best to clarify the purpose of the developed method throughout the manuscript as listed below:
In Abstract section,
line 12, we replaced “detect” to “profile”.
line 22, we deleted “accuracy and” to emphasize the selectivity.
In introduction section,
line 55, we replaced “detect” to “profile”.
Line 91, we replaced “analyze” to “profile”.
In last revision, line 54-55, we emphasized
“Therefore, establishing a rapid and selective method to profile these vicinal diols, present in trace amounts, is essential to discover their biological roles as well as for establishing new markers of diseases.” To help clarifying even further, we changed the font of “profile” and “discover”.
And in conclusion section,
Line 325 and 326, we switched the order of “selective” and “sensitive” to emphasize the selective, which is more important for profiling method.
Line 338, we added “profiling” before “analysis”.
From line 342 to 347, we added a separate paragraph to clarify that our purpose of the method is for discovering and profiling the new vicinal diols instead of quantification of them.
“It’s worth noting that the current developed method is for discovering and profiling the new vicinal diol contained compounds in the biological matrices instead of the quantification of them. For the quantification of the diols, the selected reaction monitor (SRM) mode of the triple quadrupole mass spectrometer will be much sensitive and better choice. And the SRM methods for these diols should be straightforward to develop after identification of them.”